METHODS AND RESOURCES

# STUPPIT is a proximity labeling tool for labeling intermediary proteins that bridge two non-interacting proteins

**Lin Xie**[�], **Lijuan Gao**[�], **Weihong Fu**[☣], **Gangyun Wu, Hua Li***, **Wenxiu Ning**[iD]*

Yunnan Key Laboratory of Cell Metabolism and Diseases in Yunnan Center for Life Sciences, School of Life Sciences, Yunnan University, Kunming, China

☣ These authors contributed equally to this work.
* hualee@ynu.edu.cn (HL); wenxiu_ning@ynu.edu.cn (WN)

## Abstract

Decoding the complexities of signaling pathways is fundamental for deciphering the mechanisms underlying tissue development, homeostasis, and disease pathogenesis. Proximity labeling tools have been instrumental in identifying upstream or downstream effectors of specific proteins within signaling pathways. However, currently, there are no tools available to directly label and capture intermediary proteins that bridge two non-interacting proteins. Here, we developed Split-TurboID and PUP-IT based Protein Identification Tool (STUPPIT), a novel method combining split-TurboID and PUP-IT to biotinylate intermediary proteins of two non-interacting proteins through a two-step enzymatic reaction. STUPPIT was validated using three well-characterized protein triads, including YAP1/AMOT/β-actin, YAP1/LATS1/MOB1A, and β-catenin/α-catenin/β-actin using HEK293T human cell lines. Combining STUPPIT and proteomics, we identified novel intermediary proteins including ERC1 and USP7, which interacted both with β-catenin and SMAD4, key components of the Wnt and BMP signaling pathways. In conclusion, STUPPIT represents a powerful tool for labeling and capturing intermediary proteins between non-interacting partners, offering new insights into protein-protein interactions and advancing signal transduction research.

## Introduction

Cellular signaling pathways depend on precisely regulated protein-protein interactions that form dynamic networks controlling fundamental biological processes. These interactions are not static but rather exhibit remarkable plasticity, enabling cells to adapt to microenvironments during tissue development, homeostasis, and stress responses. The identification of novel interacting partners of proteins of interest represents a crucial and central aspect of signaling pathway research [1–3].

**Data availability statement:** All relevant data are within the paper and its Supporting information files. The raw mass spectrometry (MS) data have been deposited in the public repository iProX (project ID: IPX0013559000, ProteomeXchange ID: PXD069286).

**Funding:** This study was supported by the National Natural Science Foundation of China No. 32270846, W.N.; and Applied Basic Research Foundation of Yunnan Province No. 202401AT070443, H.L. The funders had no role in study design, data collection and analysis, decision to publish, or preparation of the manuscript.

**Abbreviations:** AJ, adherens junction; APEX2, ascorbate peroxidase; FDR, false discovery rate; HRP, horseradish peroxidase; LFQ, label-free intensity quantitation; PBS, phosphate-buffered saline; STUPPIT, Split-TurboID and PUP-IT based Protein Identification Tool.

Over time, versatile tools utilizing proximal enzymatic labeling have emerged for capturing proximal partners of known proteins, such as horseradish peroxidase (HRP) [4], ascorbate peroxidase (APEX2) [5], biotin ligases (BioID [6,7] and TurboID [8]), LIPSTIC (transpeptidase sortase A) [9] or its updated version EXCELL [10], and PUP-IT (PafA ligase [11]). These proximity labeling techniques allow for the identification of proteins that have transient and weak interactions within the proximity distance of an introduced labeling enzyme [3,12,13]. The extended forms of split proximity labeling enzymes, such as split-BioID, split-APEX2, and split-TurboID, facilitate the identification of proteins between two known interacting proteins [14–16]. These proximity labeling tools have been instrumental in dissecting the upstream or downstream effectors of proteins of interest in signaling pathways, including Wnt, Hippo, Hedgehog, MAPK, and NF-κB pathways [17–22].

Despite significant advancements in the field, there remains a critical gap in our ability to directly label and capture intermediary proteins that bridge two associated but non-interacting proteins - a common scenario in signaling pathways. These intermediary proteins play a pivotal role in facilitating signal transduction and crosstalk, either by acting as adaptors to link non-interacting proteins or by competitively binding to them. For instance, in mechanotransduction, actin binds to the intermediary protein AMOT, thereby sequestering it from inhibiting YAP1 [23,24]. In adherens junctions (AJs), α-catenin serves as an intermediary protein that connects F-actin to junctional sites by interacting with β-catenin [25,26]. In the Hippo signaling pathway, MOB1 binds to the intermediary proteins LATS1/2, which in turn facilitates the phosphorylation of YAP1 by LATS1/2, blocking its nuclear import and inhibiting its transcriptional programs [27–29]. Moreover, in pathways such as Wnt and BMP, reciprocal interactions are essential for controlling stem cell activity during tissue homeostasis, as seen in the small intestine epithelium and hair follicles [30,31]. However, the mechanisms by which intermediary proteins might competitively bind to key components of these pathways to balance their activities remain poorly understood. A major reason for this gap in knowledge is the lack of tools to directly label intermediary proteins between two non-interacting but associated proteins.

To address this gap, we developed Split-TurboID and PUP-IT based Protein Identification Tool (STUPPIT), a novel proximity labeling tool specifically designed to label and capture intermediary proteins between two non-interacting proteins. STUPPIT integrates split-TurboID and PUP-IT methods to biotinylate intermediary proteins via a two-step enzymatic reaction. We utilized STUPPIT to verify known intermediary proteins for several pairs of proteins, including YAP1/AMOT/β-actin, YAP1/LATS1/MOB1A, and β-catenin/α-catenin/β-actin. All of these known intermediary proteins were successfully labeled and enriched. Furthermore, by combining STUPPIT with proteomics, we identified novel intermediary proteins, such as ERC1 and USP7, between β-catenin and SMAD4, which are key components of the Wnt and BMP signaling pathways. In conclusion, STUPPIT represents a powerful tool for labeling and capturing the intermediary proteins between two non-interacting proteins, thereby enhancing and expediting the investigation of protein-protein interactions in signaling pathways.

## Results

### STUPPIT leverages split-TurboID and PUP-IT to enable labeling of the intermediary proteins between non-interacting proteins

To label the intermediary partner that bridges two associated but non-interacting proteins, we designed the STUPPIT tool by combining the concepts of split-TurboID and PUP-IT [11,16].

Split-TurboID comprises two TurboID fragments (TbN, an N-terminal fragment, and TbC, a C-terminal fragment) split at the amino acid site L73/G74 [16]. These split fragments can be attached to the two interacting proteins and brought together to reconstitute an active enzyme, allowing the labeling of proximal partners of two known interacting proteins (Fig 1A). On the other hand, PUP-IT is a highly specific proximity labeling method based on the prokaryotic enzyme PafA, which catalyzes the phosphorylation of the C-terminal Glu on PupE and then conjugates the C-terminal Glu to the lysine residue on the bait protein and interacting prey proteins, but not distant proteins [11] (Fig 1B). Therefore, PUP-IT captured proteins can be detected by molecular weight laddering on protein gels or western blots.

Taking advantage of the flexibility of modifying the N-terminus of PupE with different tags without affecting its function, we opted to fuse the shorter fragment TbN instead of the longer TbC at the N-terminus of PupE. Notably, the labeling efficiency of the resulting TbC-PupE substrate was not as high as that of TbN-PupE (S1A Fig). The prey protein A is fused to PafA with a Myc tag. In the cellular environment with TbN-PupE present, the neighboring interacting proteins of prey A, including the intermediary protein B, will be labeled with TbN-PupE. As the intermediary protein B dissociates from prey A and interacts with the TbC-linked prey C, the combined TbN and TbC will enzymatically reactivate to label the target

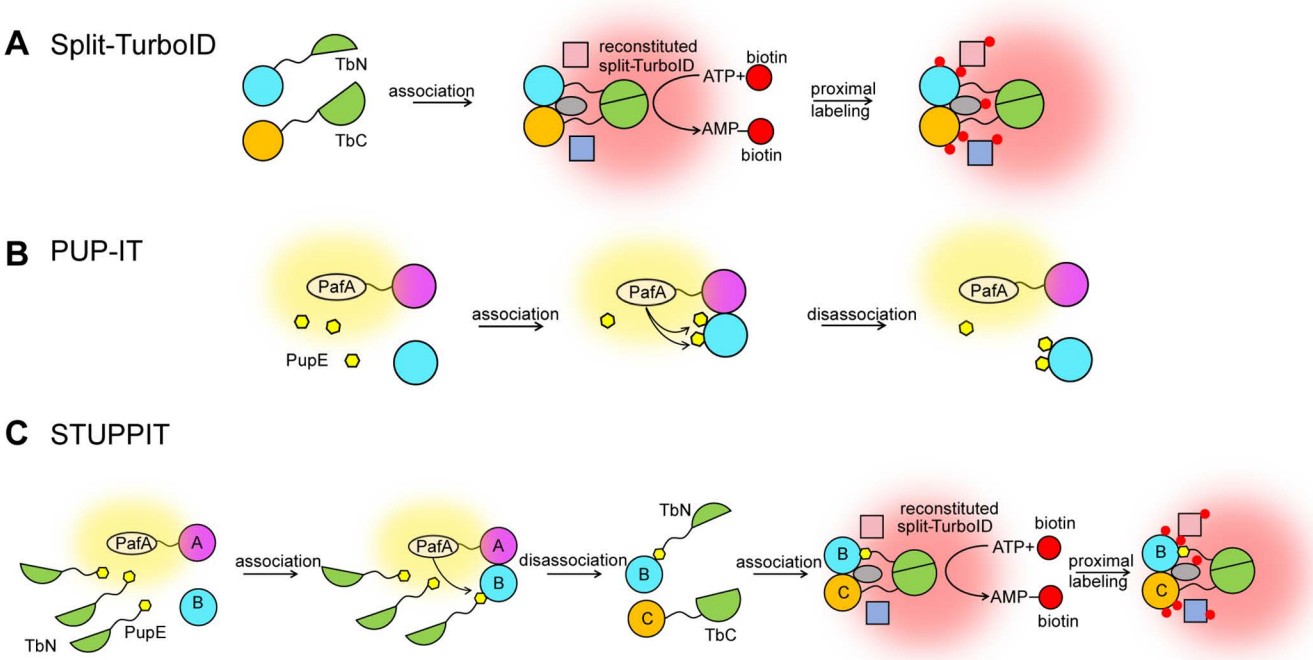

**Fig 1. Design of STUPPIT for labeling and capturing intermediary proteins between two non-interacting proteins. (A B)** Schematic diagrams depicting the working principles of split-TurboID (A) and PUP-IT (B). **(C)** Schematic representation of how STUPPIT labels the intermediary proteins between two non-interacting proteins with biotin.

proteins B and C along with their proximal proteins (Fig 1C). Hence, we named STUPPIT for this method designed to tag the intermediary connectors of two non-interacting proteins.

## Validating the intermediary protein labeling efficiency of STUPPIT by known interactors actin-AMOT-YAP1

To evaluate the feasibility of the STUPPIT concept, we developed approach I of STUPPIT by establishing HEK293T cells stably expressing 3×Flag-TbN-PupE, the substrate of PafA (Fig 2A). This approach involves the following constructs: the preyA plasmid preyA-PafA-Myc, the preyC plasmid HA-TbC-preyC, and the control HA-TbC and PafA-Myc plasmid. The stable expression of 3×Flag-TbN-PupE in HEK293T cells was validated through both immunofluorescence staining and immunoblotting (Fig 2B and 2C). We first validated approach I of STUPPIT utilizing the well-known triad, F-actin/AMOT/YAP1. The F-actin/AMOT/YAP1 axis is a well-established interaction pathway during mechanotransduction, where actin competes with YAP1 for binding to AMOT, thereby inhibiting AMOT-mediated cytoplasmic retention of YAP1 (Fig 2D) [23,24]. Thus, AMOT serves as the intermediary protein between β-actin and YAP1, and should be enriched by STUPPIT (Fig 2E). Constructs were created for β-actin-PafA-Myc, the control vector, HA-TbC and PafA-Myc, and the YAP1 vector, HA-TbC-YAP1 (Fig 2F).

We first validated whether actin-PafA can label the Flag-TbN-PupE substrate onto the intermediary protein AMOT. If this is the case, AMOT should be detectable following anti-Flag immunoprecipitation enrichment (Flag-IP) and exhibit molecular weight laddering as previously described. As expected, endogenous AMOT was successfully enriched following Flag-IP and exhibited molecular weight laddering detected by anti-AMOT in actin-PafA transfected cells, in contrast to the no-transfection or the PafA-Myc transfection control (Fig 2G). Consistent with this, a similar laddering pattern was observed for exogenous AMOT-HA following Flag-IP when co-transfected with actin-PafA (S1B Fig). Furthermore, the split-TurboID system applied to the interacting protein pair (AMOT&YAP1) generated a significantly stronger self-labeling signal than the non-interacting pair (actin&YAP1), although a low level of background self-labeling was still detectable (S1C Fig).

We further validated the efficacy of STUPPIT in labeling the intermediary protein AMOT between β-actin and YAP1 (Fig 2F). Cells stably expressing 3×Flag-TbN-PupE were co-transfected with either actin-PafA-Myc&YAP1-TbC, the actin-PafA-Myc&TbC control, or the PafA-Myc&YAP1-TbC control. These groups all showed comparable expression levels of Flag-tagged proteins, indicating similar substrate expression (Fig 2H). Importantly, AMOT exhibited molecular weight laddering and was significantly enriched in the actin&YAP1 group compared to the control groups. Together, these data demonstrate that STUPPIT uniquely enables the identification of intermediary proteins between non-interacting partners, providing access to proximal interaction information that is not available with split-TurboID alone.

## Validating intermediary protein labeling efficiency of STUPPIT by other known interactors

To further confirm the labeling efficiency of STUPPIT, we assessed its ability to label intermediary proteins in additional well-characterized protein pairs. We extended the application of STUPPIT to the MOB1A-LATS1-YAP1 cassette, which represents typical components of the Hippo pathway (Fig 3A) [27–29]. LATS1 serves as the intermediary protein between MOB1A and YAP1 (Fig 3B). Constructs were created for MOB1A-PafA-Myc, the control vector HA-TbC and PafA-Myc, and the YAP1 vector HA-TbC-YAP1 as described previously (Fig 3C). Similarly, we also detected that the intermediary protein LATS1 was successfully conjugated with 3×Flag-TbN-PupE by MOB1A-PafA. Following Flag-IP, LATS1 exhibited molecular weight laddering, in contrast to non-transfection or the PafA-Myc transfected control (Fig 3D). We also validated the efficacy of STUPPIT in labeling the intermediary protein LATS1 between MOB1A and YAP1 (Fig 3E). Consistently, LATS1 was significantly enriched in the MOB1A&YAP1 group compared to the control group, and exhibited molecular weight laddering as well.

We further extended the validation of STUPPIT by testing interactors of β-actin/α-catenin/β-catenin. The AJ, a mechanosensitive structure capable of sensing junctional tension, relies on α-catenin anchored to AJ by β-catenin [25,26]. This interaction allows α-catenin to engage with actin via its actin-binding domain, thereby facilitating the linkage of cell

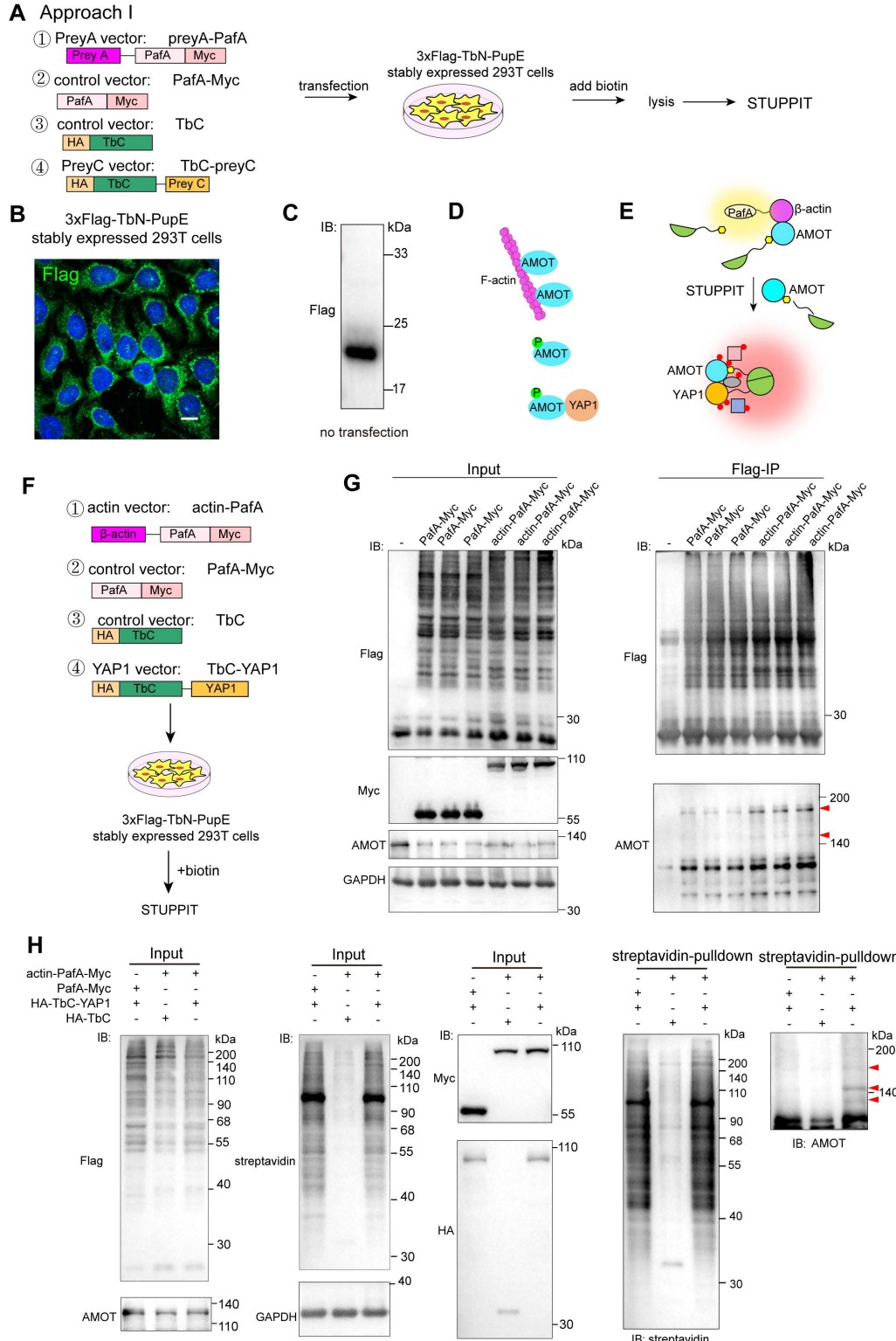

**Fig 2. Proximity labeling of the intermediary protein AMOT between β-actin and YAP1 by STUPPIT. (A)** Schematic representation of STUPPIT approach I with HEK293T cells stably expressing 3×Flag-TbN-PupE. **(B, C)** Verification of 3×Flag-TbN-PupE expression in HEK293T cells through immunofluorescence (B) and immunoblotting (C), scale bar 10 µm. **(D)** Illustration of AMOT as the intermediary protein between actin and YAP1. **(E)**

Illustration for the identification of the intermediary protein AMOT between actin and YAP1 using STUPPIT. **(F)** Schematic of the vector design for actin-PafA, TbC-YAP1, PafA and TbC control used in STUPPIT approach I. **(G)** Immunoblotting confirming that β-actin-PafA-Myc ligates the 3×Flag-TbN-PupE substrate onto endogenous AMOT, compared to PafA-Myc or non-transfected control groups. The red arrowheads indicate the molecular weight laddering of AMOT after Flag-IP. **(H)** Validation of the intermediary protein AMOT between YAP1 and β-actin captured by STUPPIT approach I through immunoblotting. 3×Flag-TbN-PupE stably expressed cells were co-transfected with actin-PafA-Myc&HA-TbC, PafA-Myc&HA-TbC-YAP1, or actin-PafA-Myc&HA-TbC-YAP1. The red arrowheads indicate the molecular weight laddering of AMOT after the streptavidin pulldown.

adhesion complexes to the actin cytoskeleton (S2A and S2B Fig) [32–34]. As anticipated, α-catenin, was also conjugated with 3×Flag-TbN-PupE by actin-PafA following anti-Flag enrichment and exhibited molecular weight laddering (S2C Fig), and was significantly enriched by STUPPIT of actin&β-catenin group compared to the control groups (S2D Fig). These results further validated the ability of STUPPIT in labeling intermediary proteins of non-interacting proteins.

## Validating intermediary protein labeling efficiency of STUPPIT by mass spectrometry analysis

We also developed the all-in-plasmid approach II of STUPPIT by validating the interactors of actin/AMOT/YAP1. Constructs were generated for prey A (actin-PafA-Myc), control (HA-TbC-IRES-3×Flag-TbN-PupE), and prey C (HA-TbC-YAP1-IRES-3×Flag-TbN-PupE) (Fig 4A). The expression of these tagged proteins was validated via immunofluorescence staining and immunoblotting (S3A–S3D Fig). 3×Flag-TbN-PupE exhibited a single band when transfected alone into cells (S3B Fig). Subsequent steps in STUPPIT involve using streptavidin affinity precipitation and mass spectrometry-based proteomics to capture and identify intermediary proteins (Fig 4A). The gradient in anti-Flag blotting and the increase in AMOT protein size by STUPPIT indicated the efficient PafA-mediated catalysis of TbN-PupE on interacting proteins as well (Fig 4B and 4C). AMOT was also enriched in the β-actin&YAP1 STUPPIT group compared to the control group (Fig 4B). We noticed that the expression of 3×Flag-TbN-PupE in the YAP1 group was less robust compared to the control group (Fig 4B). This discrepancy may be attributed to the inability of the IRES sequence to achieve 100% expression of the TbN-PupE insert, potentially compromising the efficiency of STUPPIT [35,36]. To further verify that STUPPIT can label AMOT, the intermediary protein between actin and YAP1, we enriched biotinylated proteins from two groups: the actin-PafA&TbC-YAP1 group and the actin-PafA&TbC control group (both using STUPPIT approach II), followed by mass spectrometry analysis of the samples. To enhance the efficiency of the STUPPIT system, we designed the experiment to maintain only one variable, preyC. As anticipated, compared with the control group, the actin&YAP1 STUPPIT group showed significant enrichment of the intermediary protein AMOT as well as YAP1. In contrast, LATS1, another protein that interacts with YAP1 but not actin, is not enriched (Fig 4D). We further performed STUPPIT analysis on the actin&YAP1 bait pair in both control and AMOT-knockdown stable cell lines using the all-in-plasmid method, followed by mass spectrometry (S3E Fig). Comparative analysis of the approximately 700 identified proteins revealed that about 135 proteins exhibited a significant decrease in abundance (with a fold change >1.5, calculated as control/AMOT-KD) upon AMOT knockdown (S3F Fig), implying that AMOT was indeed the main intermediary protein between actin and YAP1.

We also employed approach I of the STUPPIT system, enriching biotinylated proteins from three groups: the actin-PafA&TbC-YAP1 group, the MOB1A-PafA&TbC-YAP1 group, and the TbC-YAP1 control group (with actin-PafA excluded). The TbC-YAP control was used to identify and subtract non-specific background proteins resulting from the inherent self-ligation of split-TurboID, which produces a moderate but detectable level of self-labeling and reassembly independent of a true protein interaction (S1C Fig). These enriched biotinylated proteins were subsequently subjected to mass spectrometry-based proteomic analysis. In line with the results of approach II, the actin&YAP1 group showed enrichment of AMOT but not LATS1. Interestingly, in addition to LATS1, AMOT was also enriched in the MOB1A&YAP1 group (Fig 4E), consistent with previous reports that AMOT cooperates with MOB1A to enhance the phosphorylation of LATS1/2 [37]. This interaction between MOB1A and AMOT was further confirmed using the PUP-IT method, as evidenced by the

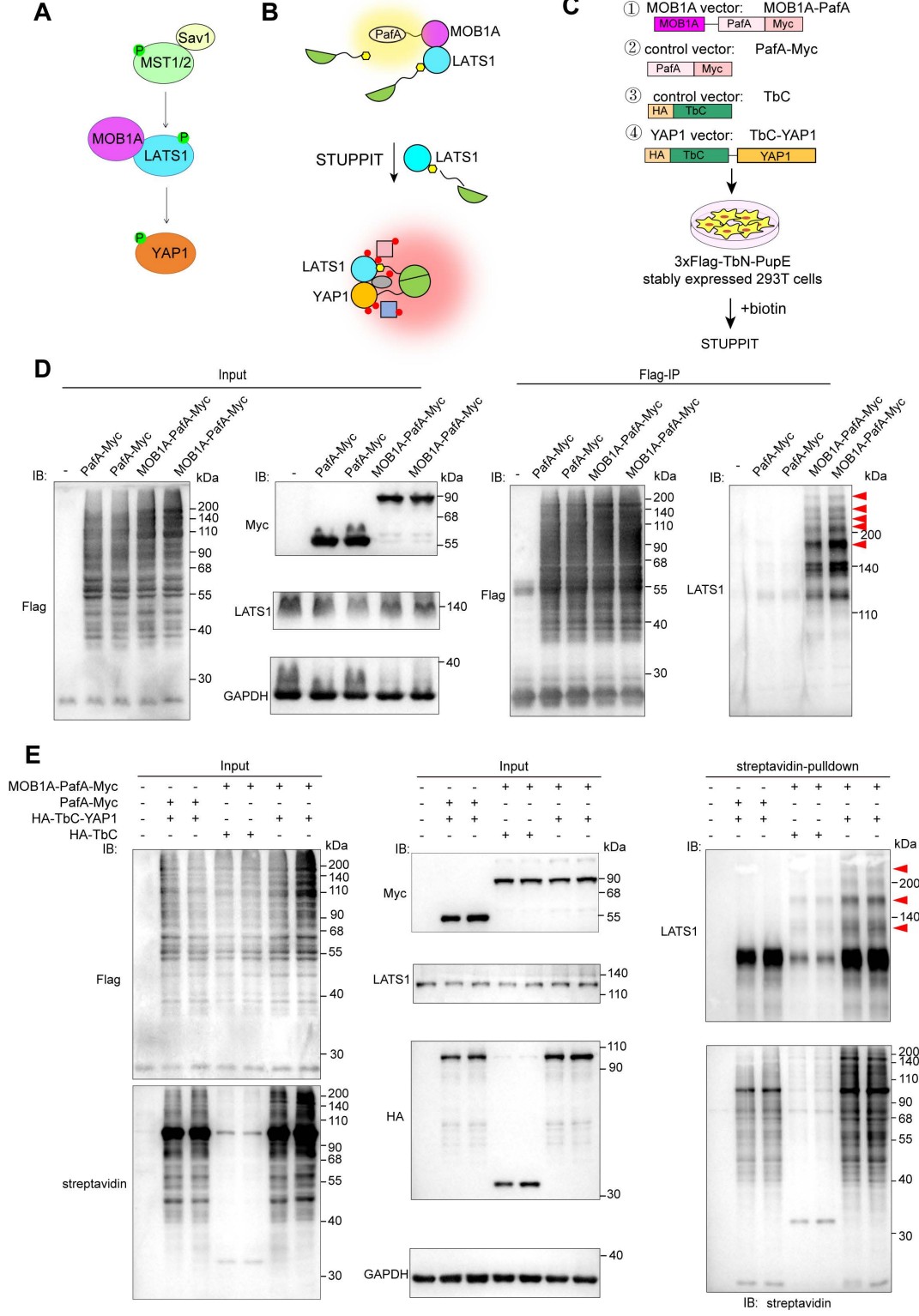

**Fig 3. Proximity labeling of the intermediary protein LATS1 between MOB1A and YAP1 by STUPPIT. (A)** Illustration of LATS1 as the intermediary protein between MOB1A and YAP1 in Hippo pathway. **(B)** Schematic of STUPPIT labeling LATS1 as the intermediary protein bridging MOB1A and YAP1. **(C)** Schematic of the vector designs for MOB1A-PafA, TbC-YAP1, PafA, and TbC control used in STUPPIT approach **I. (D)** Immunoblotting confirming

that MOB1A-PafA-Myc ligates the 3×Flag-TbN-PupE substrate onto endogenous LATS1, compared to PafA-Myc or non-transfected control groups. The red arrowheads indicate the molecular weight laddering of LATS1 after Flag-IP. **(E)** Validation of LATS1 as an intermediary protein between YAP1 and MOB1A by STUPPIT approach I. 3×Flag-TbN-PupE stably expressed cells were co-transfected with MOB1A-PafA-Myc&HA-TbC, PafA-Myc&HA-TbC-YAP1, or MOB1A-PafA-Myc&HA-TbC-YAP1. The red arrowheads indicate the molecular weight laddering of LATS1 after streptavidin pulldown.

molecular weight laddering of AMOT in MOB1A-PafA-Myc transfected 3×Flag-TbN-PupE stably expressed cells after Flag-IP (Fig 4F). These findings further underscore the effectiveness of the STUPPIT in labeling and capturing intermediary partners between two related proteins that do not have a direct interaction.

## Application of STUPPIT to identify novel intermediary proteins between proteins of interest

Finally, we sought to test whether STUPPIT can identify novel intermediary proteins between proteins of interest. We thus focused on the BMP and Wnt signaling pathways, as reciprocal interactions between these pathways are critical for regulating stem cell behaviors during tissue development and homeostasis, such as the small intestine [30] and hair follicles [31]. These two pathways are known to counteract each other, and one potential mechanism underlying this counteraction is competition for shared binding partners (Fig 5A). We therefore chose β-catenin and SMAD4 as the prey proteins for STUPPIT, as they are the core proteins of the Wnt and BMP signaling pathways, respectively (Fig 5B).

The biotinylated proteins labeled by STUPPIT were enriched from the SMAD4-PafA&TbC-β-catenin experimental group and the SMAD4-PafA&TbC control group, respectively, and were then subjected to mass spectrometry analysis (Fig 5C). The proteomics data implied that proteins such as TRIM33, RNF20, ERC1, and USP7 might be the intermediary proteins between β-catenin and SMAD4 (Fig 5D and 5E). Notably, TRIM33 has been previously reported to interact with β-catenin [38]. It also interacts strongly with SMAD3 and weakly with SMAD4 [39]. These reported interactions were further validated in our study via immunoprecipitation (S4 Fig).

We next sought to identify novel intermediary proteins between β-catenin and SMAD4 among RNF20, ERC1, and USP7. We thus generated expression constructs for these candidate proteins and validated their expression via immunofluorescence staining (S5 Fig). We examined the interactions of RNF20, ERC1, and USP7 with β-catenin and SMAD4 by Flag-IP, using HA-TbC or GFP-Flag as controls. Compared to RNF20, ERC1 and USP7 exhibited robust interactions with both β-catenin and SMAD4 (Fig 6A–6F). The PUP-IT method is a more specific, lower-background approach for labeling proximal rather than distant interacting proteins. We thus applied PUP-IT to further verify the interactions between RNF20, ERC1, USP7, and β-catenin, SMAD4. Consistently, only ERC1 and USP7 could be conjugated with Flag-TbN-PupE by both β-catenin-PafA and SMAD4-PafA (Fig 6G and 6H), whereas RNF20 could only be conjugated with Flag-TbN-PupE by β-catenin-PafA but not SMAD4-PafA (S6 Fig). These data indicate that STUPPIT identified ERC1 and USP7 as two intermediary proteins that interact with both β-catenin and SMAD4. Together, these data validated the ability of STUPPIT to label and capture the intermediary proteins between two non-interacting proteins, thereby facilitating the investigation of signaling transduction and crosstalk.

## Discussion

Identifying novel protein-protein interactions is key to understand signal transduction and crosstalk in cells and tissues. Our study introduces a novel tool, STUPPIT, which addresses a critical gap in proximity labeling by enabling the direct labeling and capture of intermediary proteins between non-interacting partners. This advancement significantly enhances our ability to dissect complex signaling pathways and their underlying mechanisms.

Proximity labeling tools such as TurboID and APEX have been instrumental in tagging endogenous proteins that interact with a specific bait protein, allowing their enrichment and identification through mass spectrometry. While split-TurboID

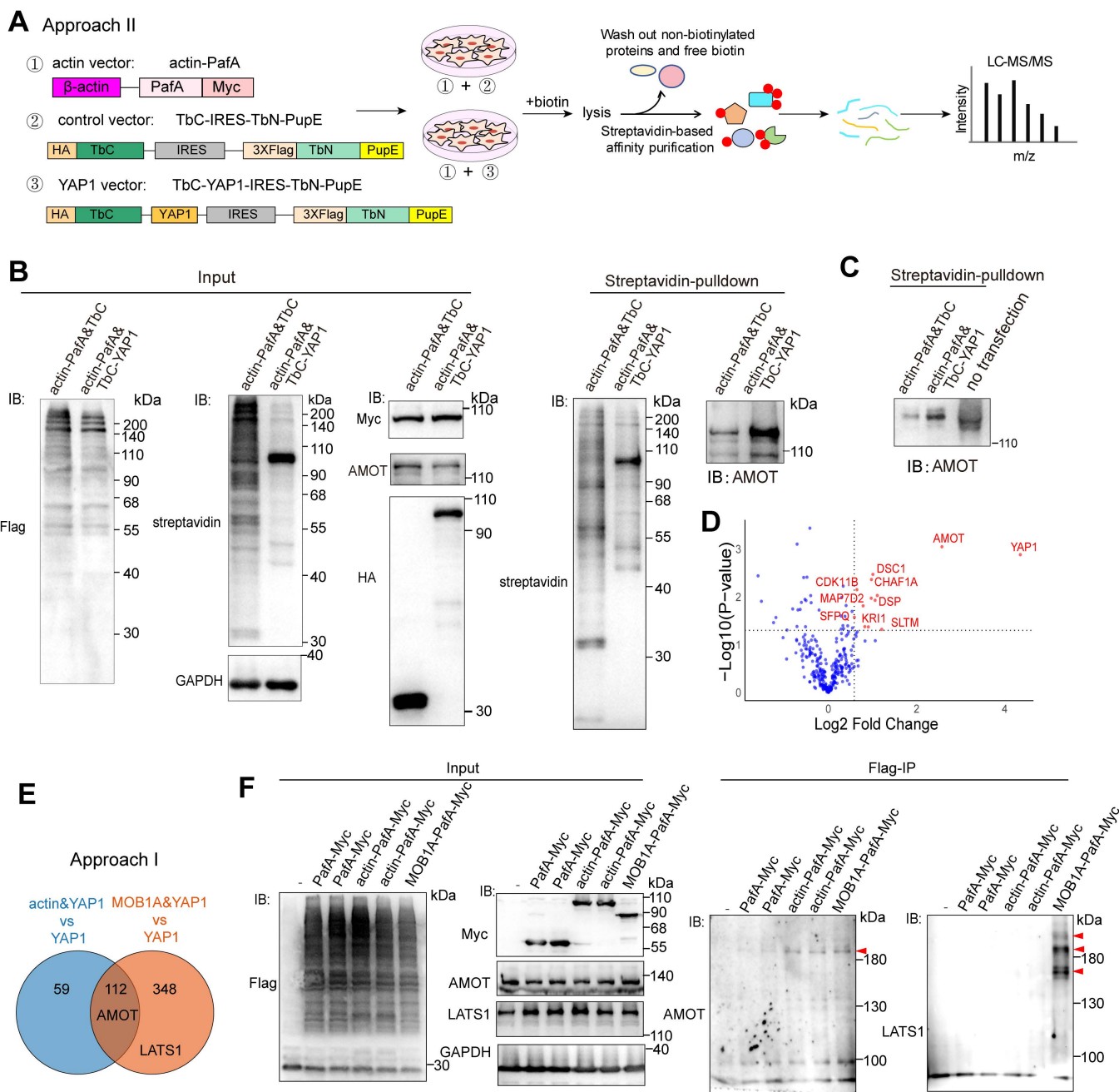

**Fig 4. Validating the intermediary protein labeling efficiency of STUPPIT by mass spectrometry analysis. (A)** Schematic of the all-in-plasmids STUPPIT approach II labeling AMOT between YAP1 and β-actin. **(B)** Immunoblotting validation of AMOT as the intermediary between YAP1 and β-actin by STUPPIT approach II. HEK293T cells were co-transfected with actin-PafA-myc&TbC-IRES-TbN-PupE or actin-PafA-myc&TbC-YAP1-IRES-TbN-PupE, respectively. The STUPPIT assay was then performed on each set of transfections. **(C)** Immunoblotting showing the enrichment and the increase in protein size of AMOT in actin-PafA&TbC-YAP1 group after STUPPIT, compared to actin-PafA&TbC or non-transfected control groups. **(D)** Volcano plots of the potential intermediary proteins between actin and YAP1 enriched by STUPPIT approach II after mass spectrometry analysis. Proteins with unique peptides > 3 in the actin-PafA&TbC-YAP1 over actin-PafA&TbC were first filtered. Proteins with > 1.5 fold change (actin-PafA&TbC-YAP1 over actin-PafA&TbC) were labeled as red in the volcano plots. **(E)** Venn diagram comparing potential intermediary proteins enriched in actin-PafA&TbC-YAP1 or MOB1A-PafA&TbC-YAP1 compared to the YAP1-TbC control, identified through STUPPIT approach I and subsequent mass spectrometry analysis. Proteins with fold change > 2 in both groups compared to the YAP1-TbC control were filtered. Specifically, AMOT was common to both groups, while LATS1 was unique to MOB1A&YAP1 group. **(F)** Immunoblotting confirming that MOB1A-PafA-Myc ligates the 3×Flag-TbN-PupE substrate onto

endogenous AMOT and LATS1. 3×Flag-TbN-PupE stably expressed cells were transfected with PafA-Myc, actin-PafA-Myc or MOB1A-PafA-Myc, respectively. The red arrowheads indicate the molecular weight laddering of AMOT or LATS1 in MOB1A-PafA-Myc transfected cells after Flag-IP.

methods can detect proximal partners of interacting proteins, they rely on close proximity to reconstitute enzymatic activity. A key limitation of these methods, however, is their inability to directly label and capture endogenous intermediary proteins between associated but non-interacting proteins. While performing separate proximity-labeling experiments on proteins A and B, followed by cross-comparison can partially address this gap, this strategy remains indirect and suffers from several drawbacks. For instance, candidate intermediary proteins must not only be efficiently labeled by both baits but also pass two independent background filtering steps. Moreover, this approach demands increased sample input and needs higher experimental costs. In contrast, STUPPIT offers a solution by enabling the specific direct labeling and enrichment of intermediary proteins between two non-interacting bait proteins. This unique capability positions STUPPIT as a valuable alternative tool for investigating signaling pathways, particularly those where key components lack direct physical interactions.

It is worth noting that other split-proximity labeling technologies such as split-APEX2 [40] or split-HaloTag system [41] could theoretically replace split-TurboID in the STUPPIT system for labeling intermediary proteins in cells. However, the length of the split-fragment linked to PupE must be carefully optimized to ensure efficient tagging of the modified PupE to the lysine amino acid in the intermediary proteins by PafA. Furthermore, to replace split-TurboID in STUPPIT, split-X must exhibit high specificity and low background noise when labeling the interacting proteins. This optimization is critical for maintaining the specificity and efficiency of the labeling process. A limitation of STUPPIT is that both PafA and TurboID target lysine residues for modification. The initial PUP-IT step can thus saturate these sites, precluding subsequent biotinylation by TurboID. A future enhancement could therefore involve integrating PUP-IT with a complementary system like split-APEX2, which biotinylates tyrosines to avoid this competition. Notably, our system establishes a conceptual framework: combining different labeling tools or biochemical enzymes, including proximity labeling enzymes or even dCas9 (for transcriptional level studies), could expand the potential for targeted labeling or functional interrogation.

Moreover, STUPPIT not only labels intermediary proteins and prey C but also identifies proximity proteins between the intermediary protein and prey C. This dual functionality is advantageous, as it further enriches the core components of intermediary proteins and prey C proteins, facilitating subsequent investigations into their relationship. This also implies that STUPPIT can be used as an alternative to split-TurboID for labeling proximal proteins between two interacting proteins. The successful identification of intermediary proteins between SMAD4 and β-catenin, key components of the BMP and Wnt pathways, highlights the broad application of STUPPIT in exploring the crosstalk between signaling pathways. These intermediary proteins, such as ERC1 and USP7, interact both with SMAD4 and β-catenin, implying potential competitive binding between these pathways. This finding underscores the potential of STUPPIT to reveal novel regulatory mechanisms in signaling networks.

One limitation shared by STUPPIT, similar to split-TurboID, is the presence of background noise in control group. To mitigate this issue, it is advisable to maintain only one variable in the STUPPIT system between two prey proteins (such as using preyA-PafA&TbC or PafA&preyC-TbC as control). Furthermore, we recommend maintaining as uniform expression as possible for all bait proteins and TbN-PupE substrates. To achieve this, we suggest first establishing cell lines stably expressing TbN-PupE, followed by performing STUPPIT approach I. This strategy is preferred because the Flag-TbN-PupE substrate expression is more uniform in stable cell lines. In contrast, STUPPIT approach II (the all-in-plasmid method) relies on the transient co-transfection of all bait plasmids and substrate plasmids. This transient system often leads to unstable and uneven protein expression across samples. Additionally, the internal ribosome entry site (IRES) sequence used in the all-in-one plasmid of approach II does not drive fully efficient translation

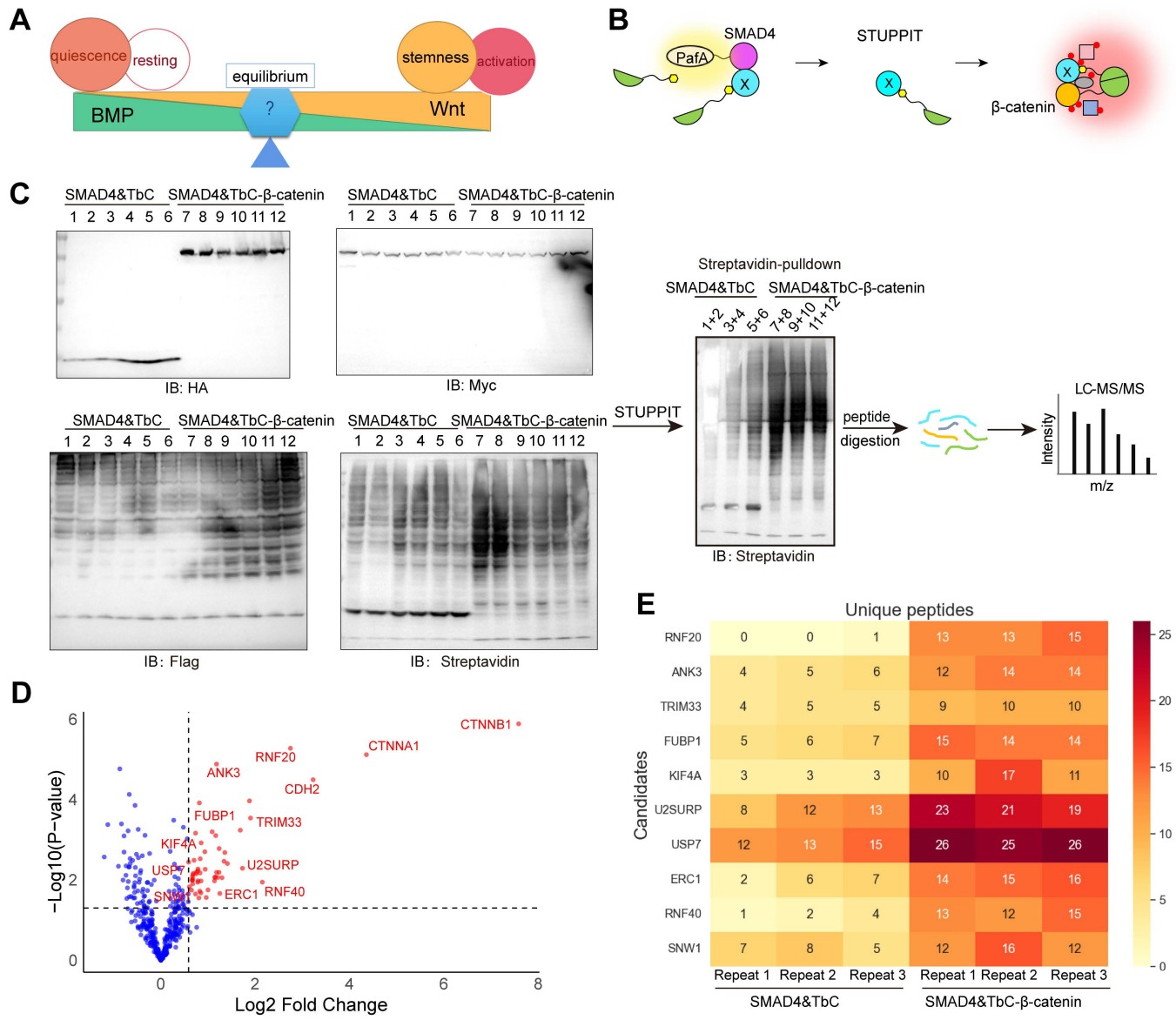

**Fig 5. Application of STUPPIT to identify novel intermediary proteins between SMAD4 and β-catenin, central hubs of BMP and Wnt pathway.**
**(A)** Schematic of the opposing effects of BMP and Wnt signaling on stem cell behavior. **(B)** STUPPIT strategy for labeling proteins that bridge SMAD4 and β-catenin. **(C)** STUPPIT workflow for capturing intermediary proteins between SMAD4 and β-catenin. 3×Flag-TbN-PupE stably expressed cells were co-transfected with SMAD4-PafA-Myc&HA-TbC or SMAD4-PafA-Myc&HA-TbC-β-catenin. STUPPIT labeled intermediary proteins between SMAD4 and β-catenin were enriched by streptavidin pulldown and analyzed by mass spectrometry. Samples 1–6 (SMAD4&TbC) and 7–12 (SMAD4&TbC-β-catenin) were each pooled into three groups (1+2, 3+4, 5+6) and (7+8, 9+10, 11+12) to yield three replicates per condition. **(D)** Volcano plots of the potential intermediary proteins between SMAD4 and β-catenin enriched by STUPPIT approach I after mass spectrometry analysis. Proteins with unique peptides ≥10 in the SMAD4&TbC-β-catenin over SMAD4&TbC control group were first filtered. Proteins with fold change (SMAD4&TbC-β-catenin over SMAD4&TbC) > 1.5 were labeled as red dots in the volcano plots. **(E)** Heatmap of the unique peptide counts for the indicated candidates in SMAD4&β-catenin and SMAD4&control groups identified by mass spectrometry analysis. Underlying mass spectrometry data can be found in the ProteomeXchange (PXD069286).

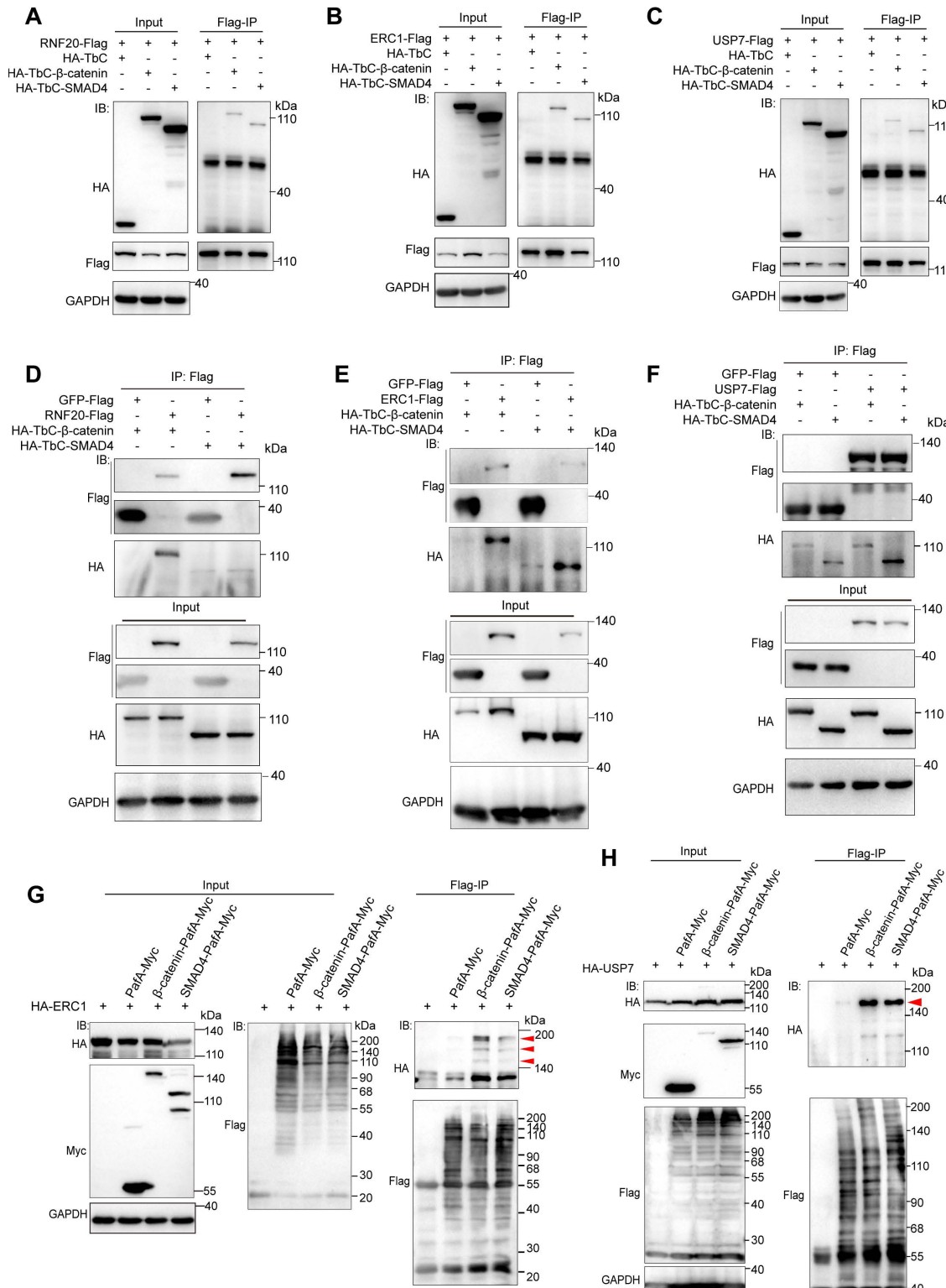

**Fig 6. Validation of the intermediary proteins between β-catenin and SMAD4. (A–F)** Immunoprecipitation assays showing the interactions between RNF20 (A, D), ERC1 (B, E), and USP7 (C, F) with SMAD4 and β-catenin, respectively, as validated by Flag-IP. In these assays, HA-TbC was used as the control for panels A–C, while GFP-Flag was used as the control for panels D–F. **(G, H)** PUP-IT confirmed ERC1 (G) and USP7 (H) interact with

β-catenin and SMAD4, respectively. The immunoblots show that β-catenin-PafA-Myc and SMAD4-PafA-Myc ligate 3 × Flag-TbN-PupE substrate to HA-ERC1 (G), and HA-USP7 (H), respectively. The red arrowheads indicate the molecular weight laddering of HA-ERC1 and HA-USP7 protein after Flag-IP, respectively.

of the downstream TbN-PupE insert, which could compromise the efficiency and sensitivity of the STUPPIT. Furthermore, when identifying true intermediary proteins following STUPPIT and mass spectrometry analysis, it is essential to consider the background labeling in the control group. Such analyses should be taken into consideration to narrow down potential candidates. A more targeted approach for identifying intermediary proteins with STUPPIT is to employ sequential Flag immunoprecipitation followed by streptavidin pull-down prior to mass spectrometry. This two-step purification strategy can ensure the highly specific enrichment of double-labeled proteins, resulting in a list of high-confidence candidates. Subsequent validation steps, such as immunoprecipitation by Flag-IP or PUP-IT, are necessary to confirm the true intermediary proteins.

Future work should focus on optimizing the STUPPIT system to reduce background noise and improve labeling efficiency. Additionally, exploring the application of STUPPIT in different cell types and tissues will provide broader insights into the dynamic protein-protein interactions in various biological contexts. The intermediary proteins uncovered using STUPPIT may reveal previously undruggable nodes in signaling networks, offering new therapeutic entry points and deeper mechanistic insight into health and disease.

## Materials and methods

### Lead contact

Any requests for further information and resources should be directed to Wenxiu Ning (wenxiu_ning@ynu.edu.cn), the lead contact on this paper.

### Materials availability

New plasmids generated in this study are available from Dr. Ning.

### Experimental model and subject details

**Cells.** All cells were cultured with a DMEM medium (Thermo Fisher, #12634-010, US) containing 10% fetal bovine serum (Vivacell#C04001-500, China) and 1% penicillin-streptomycin (Beyotime, #C0222, China) at 37 °C with 5% $CO_2$. HEK293T cells were used for immunoprecipitation assays. HeLa cells were used for immunofluorescence staining. DNA transfection of HEK293T and HeLa cells was performed by the HighGene plus transfection reagent (ABclonal, China). To establish 3 × Flag-TbN-PupE stably expressed HEK293T cells, psPAX2, pMD2G, pPHAGE-EF1α-puro-3 × Flag-TbN-PupE plasmids, and HEK293T cells were used for virus generation with PEI reagent (Polysciences#24765, US). The produced packaging lentiviruses were then added to the HEK293T cells with 10 μg/mL polybrene (Biosharp#BL628A, China), and incubated for 48 hours, after which cells were selected with 1 μg/mL puromycin for 3 days.

**Plasmids construction.** PafA-Myc DNA fragment and PupE peptide were amplified from plasmids pEF6a-kozak-CD28-PafA-Myc and pEF6a-HB-PupE provided by Prof. Min Zhuang (ShanghaiTech University). IRES DNA fragment was amplified from plasmid pLVX-mgSrtA-IRES-ZsGreen provided by Prof. Peng Chen (Peking University). TbN and TbC DNA fragments (split at the amino acid site L73/G74) were amplified from a TurboID plasmid pLVX-Flag-TurboID provided by Prof. Wenxiang Fu (Yunnan University). YAP1, β-actin, MOB1A, β-catenin, RNF20, ERC1, USP7, and TRIM33 were amplified from cDNA of human HaCaT keratinocytes, respectively. pCMV-Tag2B plasmid was provided by Prof. Jianwei Sun. pHAGE-EF1a-puro was provided by Prof. Maorong Chen from Yunnan University. Self-made plasmids

including pCMV-actin-PafA-Myc, pCMV-HA-TbC-YAP1, pCMV-HA-TbC, pCMV-HA-TbC-IRES-3×Flag-TbN-PupE, pCMV-HA-TbC-YAP1-IRES-3×Flag-TbN-PupE, pPHAGE-EF1A-puro-3×Flag-TbN-PupE are available upon request from the Lead Contact.

We designed the infusion cloning primers using Vazyme's CE Design (https://www.vazyme.com) and performed the plasmid recombination with ClonExpress II One Step Cloning Kit (Vazyme, #C112-01, China) following the manufacturer's description.

**Cell transfection.** Cells were seeded at $3 \times 10^6$ cells per 6 cm plate to achieve approximately 50% confluence. The following day, transfection was performed following protocols provided for HighGene reagents (Abclonal, #RM09014P, China). First, the culture medium was replaced with fresh DMEM. In a centrifuge tube, 400 μL of DMEM medium was added, followed by 4 μg of plasmid DNA, and then 8 μL of HighGene transfection reagent. The mixture was gently vortexed to ensure thorough mixing and then incubated at room temperature for 15 min. The transfection mixture was then evenly and gently added to each dish, and the dish was gently shaken to ensure even distribution of the mixture over the cells. Finally, the dishes were placed in the incubator for cell culture, allowing the cells to uptake the exogenous DNA.

**Cell lysis.** Cells were washed twice with phosphate-buffered saline (PBS) and lysed with an appropriate volume of RIPA buffer (400 μL for a 6 cm dish, 1 mL for a 10 cm dish). After scraping and incubate for 10 min on ice, the lysates were centrifuged at 13,000$g$ for 10 min. The supernatants were collected, supplemented with PMSF at a ratio of 1:200. Aliquot a portion of the lysate and mix it with 3×sample buffer for protein analysis before ultrafiltration. The remainder was processed for ultrafiltration. The RIPA buffer (100 mL) was prepared as follows: 3 mL 5M NaCl, 1 mL 0.5 M EDTA, pH 8.0, 5 mL 1M Tris-HCl, pH 8.0, 1 mL NP-40, 5 mL 10% sodium deoxycholate, 10% SDS: 1 mL, and ddH$_2$O to bring the volume to 100 mL.

**Ultrafiltration to remove free biotin.** The prepared lysate was transferred to a pre-chilled ultrafiltration column (Pall Microsep MCP003C4-5 mL). The volume was adjusted to 4 mL with PBS containing PMSF (1:200) and centrifuged at 4,000$g$ for 1–1.5 hours. The washing step was repeated three times to ensure complete removal of free biotin (Sigma, #B4501, US). The retentate was concentrated to about 400 μL after the final centrifugation. An 80 μL aliquot was mixed with 40 μL of 3×sample buffer as the input control. The remainder was used for streptavidin affinity immunoprecipitation.

**Streptavidin affinity pulldown assays.** Streptavidin agarose resin (YEASEN, #20512ES08, China) was equilibrated by washing three times with 1 mL of 1×TBS. The required bead volume (15 μL for a 6 cm tube, and 30 μL for a 10 cm tube) was transferred to pre-chilled lysates and adjusted to 1 mL with PBS. Samples were incubated with rotation at room temperature for 2 hours. Beads were pelleted and washed 3–4 times with 1 mL PBS or PBST (with 0.05% Tween-20). Bound proteins were eluted in 1× sample buffer by heating at 95 °C for 10 min. The 3× Sample Buffer (85 mL) was made as follows: 18.8 mL 1M Tris pH 6.8, 10.0 mL 20% SDS, 30.0 mL glycerol, 10.0 mL 0.4% bromophenol blue solution, 16.2 mL H$_2$O. Add β-mercaptoethanol or 15 mM DTT to individual aliquots when using (850 μL 3× sample buffer + 150 μL BME).

**Immunoprecipitation of Flag fusion proteins.** HEK293T cells were transfected with plasmids expressing Flag-fused proteins. After 48 hours, cells were washed with PBS and lysed in 500 μL of ice-cold RIPA buffer supplemented with PMSF, and protease inhibitors. The lysates were centrifuged at 12,000 rpm for 5 min at 4 °C. A 100 μL aliquot of the supernatant was saved as input. Anti-Flag affinity beads (Beyotime, #P2282, China) were equilibrated by washing three times with 1× TBS. For each 6 cm dish, 8 μL of beads were added to the lysates and incubated at 4 °C for 2 hours or overnight. The beads were then collected by centrifugation and washed three times with 500 μL of 1× PBS. Bound proteins were eluted in 60 μL of 1× sample buffer by heating at 95°C for 5 min.

**Western blotting.** Samples were separated by SDS-PAGE and transferred onto PVDF membranes. The membranes were blocked with 5% nonfat milk for 1 hour at room temperature and subsequently incubated with primary antibody diluted in 2% nonfat milk for 2.5 hours. After three washes with PBST, the membranes were probed with HRP-conjugated secondary antibody diluted with 2% nonfat milk for 1 hour at RT. Following another series of washes with PBST, the protein signals were detected using a chemiluminescence imaging system. Antibodies used were listed as follows: Horse

Anti-mouse IgG, HRP-linked Antibody, CST#7076; Goat Anti-rabbit IgG, HRP-linked Antibody, CST#7074; HRP-labeled Streptavidin, Beyotime#A0303; Rb anti Alpha E-Catenin, proteintech#12831-1-AP; Rb anti-AMOT, proteintech#24550-1-AP; ms anti-Flag M2, Sigma#F1804; ms anti-HA tag, Invitrogen#26183; Rb anti-LATS1, proteintech#17409-1-AP; ms anti-Myc, abclonal#AE010.

**Immunofluorescence staining.** Cells were seeded on glass-coverslips in 24-well plates at a density of $1 \times 10^5$ cells per well and transfected with the indicated plasmids. After 48 hours, the cells were fixed with 4% paraformaldehyde for 10 min and permeabilized with 0.1% Triton X-100 in PBS during three 5-minute washes. The samples were then blocked with 1% BSA buffer for 30 min, followed by incubation with primary antibodies for 1 hour. After another series of washes, the cells were incubated with fluorescently labeled secondary antibodies and DAPI for 1 hour. Finally, the coverslips were washed, mounted onto glass slides with an anti-fade mounting medium, and imaged.

**Imaging.** Slides were imaged using an inverted Zeiss LSM800 confocal microscope with Airyscan using 63× oil-immersion objective lens and processed using Fiji image.

**Mass spectrometry sample preparation and data analysis.** First, STUPPIT streptavidin affinity precipitation was conducted to purify the biotinylated protein samples. Subsequently, the purified biotinylated protein samples were first run in a 10% SDS-PAGE gel, followed by excision of the entire protein bands. In-gel digestion and LC–MS/MS analysis were performed following the protocol as described [42]. For Figs 4E and S3F, two replicates for each group were performed and sent to Chi Biotech Co. for mass spectrometry analysis; protein identification and label-free intensity quantitation (LFQ) were performed by PEAKs Studio version 10.6. For Figs 4D and 5D, 3 replicates for each group were performed and submitted to the mass spectrometry facility of Yunnan University for mass spectrometry analysis. Protein identification and LFQ were performed by MaxQuant software (version 1.6.4.0) [43]. Missing values were imputed by random numbers from a normal distribution applying a downshift of 1.8 times the standard deviation of the global dataset, and a width of 0.3 times the standard deviation. Perseus was used to calculate enrichments (Fisher's exact test) and $p$-values were presented as false discovery rate (FDR)-adjusted (FDR < 0.05). The raw mass spectrometry (MS) data have been deposited in the public repository iProX [44,45] (project ID: IPX0013559000, ProteomeXchange ID: PXD069286).

**Quantifications and statistical analysis.** Prior to performing statistical tests, Normality of data distribution was assessed by D'Agostino-Pearson test using GraphPad Prism 5 software. Data are represented as the mean ± SEM, and were judged to be statistically significant when $p$-value < 0.05 by unpaired Student t test (ns = not significant, * $p < 0.05$; ** $p < 0.01$, *** $p < 0.001$, **** $p < 0.0001$). All data were performed at least three independent replicates. All images were analyzed using FIJI ImageJ. Statistical analysis was performed using GraphPad Prism 5 software and Microsoft Excel.

## Supporting information

**S1 Fig. Immunoblotting confirmation that β-actin-PafA-Myc ligates 3 × Flag-TbN-PupE substrate to HA-AMOT. (A)** Validation of the labeling efficiency of TbC-PupE and TbN-PupE on interacting proteins by PafA. **(B)** Immunoblotting confirming that actin-PafA-Myc ligates the 3 × Flag-TbN-PupE substrate onto exogenous AMOT-HA. 3 × Flag-TbN-PupE stably expressed cells were transfected with AMOT-HA alone, PafA-Myc&AMOT-HA, or actin-PafA-Myc&AMOT-HA. The red arrowhead indicates the molecular weight laddering of AMOT-HA after Flag-IP. **(C)** Split-TurboID between actin-Flag-TbN and HA-TbC-YAP1, AMOT-Flag-TbN and HA-TbC-YAP1. The red arrowheads indicate size of AMOT-Flag-TbN, and blue arrowheads indicate size of HA-TbC-YAP1, while asterisks represent non-specific bands.
(TIF)

**S2 Fig. Proximity labeling of the intermediary protein α-catenin between actin and β-catenin by STUPPIT. (A)** α-catenin serves as the intermediary protein linking β-catenin and actin. **(B)** Schematic representation of the labeling of the intermediary protein α-catenin between actin and β-catenin using STUPPIT. **(C)** Validation of the conjugation of substrate 3 × Flag-TbN-PupE to α-catenin-HA by actin-PafA-Myc through immunoblotting. 3 × Flag-TbN-PupE stably

expressed cells were transfected with α-catenin-HA&PafA-Myc, or α-catenin-HA&actin-PafA-Myc. The red solid arrows indicate the molecular weight laddering of α-catenin-HA in actin-PafA-Myc transfected cells after Flag-IP. **(D)** Validation of the intermediary protein α-catenin between β-catenin and actin captured by STUPPIT through immunoblotting. 3 × Flag-TbN-PupE stably expressed cells were transfected with α-catenin-V5&PafA-Myc&TbC-β-catenin, α-catenin-V5&actin-PafA-Myc&HA-TbC, or α-catenin-V5&actin-PafA-Myc&TbC-β-catenin. The red solid arrow indicates the molecular weight laddering of α-catenin-V5 after the streptavidin-pulldown.
(TIF)

**S3 Fig.   Validation of the YAP1 and actin related plasmids expression in the approach II of STUPPIT. (A)** Immuno-fluorescence staining of Flag and HA tag in HA-TbC-IRES-3 × Flag-TbN-PupE control vector transfected HeLa Cells. **(B)** Immunoblotting of Flag in HA-TbC-IRES-3 × Flag-TbN-PupE expressed HEK293T cells. **(C)** Immunofluorescence staining of HA tag in HA-TbC-YAP1-IRES-3 × Flag-TbN-PupE transfected HeLa cells. **(D)** Immunofluorescence staining of Myc tag in actin-PafA-Myc transfected HeLa Cells. Scale bars are all 10 μm. **(E)** Immunoblotting of AMOT and GAPDH in control and AMOT-KD cells. **(F)** Decreased proteins captured by STUPPIT of actin&YAP1 after AMOT knockdown compared to control after mass spectrometry using the all-in-plasmid method. Two replicates for each group were performed. Around 135 proteins among the approximately 700 proteins showed decreased abundance (with a fold change > 1.5, calculated as control/AMOT-KD) upon AMOT knockdown.
(TIF)

**S4 Fig.   Analysis of TRIM33 interactions with β-catenin and SMAD proteins. (A)** Validation of the interaction between TRIM33 with β-catenin and SMAD4 captured by STUPPIT through immunoblotting. **(B)** The interaction between TRIM33 and β-catenin with or without Wnt3a stimulation by Flag-IP. **(C)** Validation of the interaction between TRIM33 with SMAD1, SMAD2, SMAD3, SMAD4, SMAD5 via immunoblotting following Flag-IP. **(D)** Validation of the interaction between TRIM33 with β-catenin and SMAD3 via immunoblotting following Flag-IP.
(TIF)

**S5 Fig.   Immunofluorescence analysis of the indicated proteins expressed in HeLa cells.**
(TIF)

**S6 Fig.   Validation of the interaction of RNF20 with β-catenin and SMAD4 using PUP-IT system.** The immunoblots show that β-catenin-PafA-Myc but not SMAD4-PafA-Myc ligates 3 × Flag-TbN-PupE substrate to RNF20-HA. 3 × Flag-TbN-PupE stably expressed HEK293T cells were co-transfected with RNF20-HA and PafA-Myc, β-catenin-PafA-Myc or SMAD4-PafA-Myc, respectively. The red arrowheads indicate the molecular weight laddering of RNF20-HA after Flag-IP.
(TIF)

**S1 Raw Images.   PDF file containing un-cropped images of all western blots in this manuscript.**
(PDF)

## Acknowledgments

We are grateful to Professor Xuna Wu for mass spectrometry analysis, the core facilities of Yunnan University for imaging, Professor Wenxiang Fu, Maorong Chen, Jianwei Sun from Yunnan University, Min Zhuang from ShanghaiTech University, and Peng Chen from Peking University for kindly gifts of plasmids as indicated in methods.

## Author contributions

**Conceptualization:** Hua Li, Wenxiu Ning.

**Formal analysis:** Lin Xie, Hua Li, Wenxiu Ning.

**Funding acquisition:** Hua Li, Wenxiu Ning.

**Investigation:** Lin Xie, Lijuan Gao, Weihong Fu, Hua Li.

**Methodology:** Lin Xie, Lijuan Gao, Weihong Fu, Hua Li.

**Resources:** Hua Li, Wenxiu Ning.

**Supervision:** Hua Li, Wenxiu Ning.

**Validation:** Lin Xie, Lijuan Gao, Weihong Fu, Gangyun Wu, Hua Li.

**Visualization:** Lin Xie, Lijuan Gao, Weihong Fu, Hua Li.

**Writing – original draft:** Hua Li, Wenxiu Ning.

**Writing – review & editing:** Hua Li, Wenxiu Ning.

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
