## [Editor Report · Decision Letter 0]

22 May 2025

Dear Dr Ning,

Thank you for submitting your manuscript entitled "A novel tool for labeling intermediary proteins between two non-interacting proteins" for consideration as a Methods and Resources Article by PLOS Biology.

Your manuscript has now been evaluated by the PLOS Biology editorial staff, as well as by an academic editor with relevant expertise, and I am writing to let you know that we would like to send your submission out for external peer review.

Once your full submission is complete, your paper will undergo a series of checks in preparation for peer review. After your manuscript has passed the checks it will be sent out for review. To provide the metadata for your submission, please Login to Editorial Manager (https://www.editorialmanager.com/pbiology) within two working days, i.e. by May 24 2025 11:59PM.

Kind regards,

Richard

Richard Hodge, PhD

rhodge@plos.org

PLOS

---

## [Decision Letter · Decision Letter 1]

7 Jul 2025

Dear Dr Ning,

Thank you for your patience while your manuscript "A novel tool for labeling intermediary proteins between two non-interacting proteins" was peer-reviewed at PLOS Biology as a Methods and Resources article. Please accept my sincere apologies for the delays that you have experienced during the peer review process. Your manuscript has now been evaluated by the PLOS Biology editors, an Academic Editor with relevant expertise, and by two independent reviewers.

In light of the reviews, which you will find at the end of this email, we would like to invite you to revise the work to thoroughly address the reviewers' reports.

As you will see, both reviewers are generally positive and think the STUPPIT approach is interesting and well designed. Reviewer #1 asks for additional clarification about the advantages offered by the system and the rationale for selecting Split-TurboID specifically. Reviewer #2 notes that additional control experiments are needed to fully validate and demonstrate utility and both reviewers raise concerns with the quality/presentation of the Western blot data. In addition, Reviewer #2 requests that an additional control experiment is provided whereby Split-TurboID experiments are conducted with baits A and C, to confirm that Split-TurboID is limited to proteins that physically interact directly.

Given the extent of revision needed, we cannot make a decision about publication until we have seen the revised manuscript and your response to the reviewers' comments. Your revised manuscript is likely to be sent for further evaluation by all or a subset of the reviewers.

**IMPORTANT - SUBMITTING YOUR REVISION**

*Re-submission Checklist*

*Published Peer Review*

*PLOS Data Policy*

*Blot and Gel Data Policy*

Best regards,

Richard

Richard Hodge, PhD

rhodge@plos.org

REVIEWS:

Reviewer #1: Sequential protein-protein interaction relays are critical for signal transduction, making the capture of these interactions essential. In this study, Xie et al. developed STUPPIT, a proximity labeling tool combining PUP-IT with split TurboID to identify network components when two distantly related proteins are known. The system is well-designed: the authors validated its feasibility using established PPI networks and applied it to map new components in the SMAD pathway. While validation of the newly identified intermediary protein between SMAD4 and catenin remains incomplete, this methodological study establishes a foundation for future investigations.

A few minor points need clarification:

1. What advantages does STUPPIT offer for identifying intermediate interactors between proteins A and B compared to performing proximity labeling on each protein individually and then cross-comparing results to find intermediate interacting proteins? Please elaborate on this comparison in the discussion.

2. In Fig. 5E-H, proteins with identical tags should be presented in a single blot rather than separated by molecular weight. The current presentation obscures critical information about relative expression levels between control and test samples. In Figs. 3 & 4, the stronger biotin signals in control samples suggest that split TurboID exhibits high background when overexpressed in cells. Therefore, control Tb(C) and fused Tb(C) must be expressed at similar levels.

3. Why do RNF20, ERC1, TRIM33, and USP7 display different cellular localizations despite interacting with the same protein?

4. PUPIT uniquely uses a peptide substrate for method development, but numerous split systems exist beyond split TurboID, including split APEX and split BioID. What is the rationale for selecting TurboID, and how broadly applicable is this system?

5. Fig 2A, 3A, should be "approach"

6. In Fig 3D, lanes (2) and (3) are labeled "IB:Myc" but should likely be "IB:HA."

7. Fig 5C: clarify the meaning of the numbers above the Western blot bands and provide additional detail in the figure legend.

Reviewer #2: In this manuscript Xie L. et al. present an interesting method to study protein-protein interactions that aims at addressing the question: if considering protein A and protein C that are not physically directly interacting with one another, is it possible to identify proteins that interact with C after a prior interaction with A. Such potential proteins are dubbed "intermediary proteins" and may be especially relevant in signalling cascades.

The method is based on a clever setup that combines the PUP-IT and split-turboID techniques in an approach termed STUPPIT. In PUP-IT, the PafA protein ligase is expressed in mammalian cells in presence of its protein substrate Pup(E), PafA then non-specifically conjugates Pup(E) on lysine residues of any nearby protein. In split-turboID, two complementary fragments of the protein biotin ligase turboID are fused to two interacting proteins that trigger re-assembly of the active turboID enzyme, leading to non-specific biotinylation of neighbouring proteins.

In STUPPIT, a first bait protein (A) is fused with PafA: Prey A-PafA. In contrast to PUP-IT, the subtrate for PafA is however a fusion protein made of one fragment of split-turboID fused with pup(E): Tb(N)-pup(E). Then a second bait protein (C) is fused with the complementary split-turboID fragment: Tb(C)-Prey C. The idea is that if a protein interacts with both Prey A and Prey C (be it simultaneously or sequentially), it will first be conjugated with Tb(N)-pup(E) because of the interaction with Prey A and then activate turboID-mediated biotinylation due to the interaction with Prey C and re-assembly of the Tb(C) and Tb(N) fragments.

The manuscript is generally clear and the experiments well-conceived. The method is potentially interesting but to my opinion some more control experiments are needed to demonstrate that it is really filling a missing spot in the toolbox of methods available for the study of protein-protein interactions. In addition the description of the method is several times ambiguous as to its capacity to identify intermediary proteins or solely proteins that are parts of a complex that contains C in addition to a protein that previously interacted with A (both are not the same!)

One easy additional experiment that the authors should perform to demonstrate the utility of their method is performing split-turboID experiments with all the baits A and C used in this study. The authors write that split-turboID and related-techniques are limited to proteins that physically interact directly, this was to my knowledge often claimed but never really demonstrated: physically what matters is if the linkers allows the re-assembly of the two halves within a complex, if this automatically correlates with the necessity to be fused to two directly interacting proteins does not seem to be something to take for granted. If the authors do not observe biotinylation for all their baits, then they would show that STUPPIT brings something more than split-turboID.

Here are more detailed comments to the authors:

1)

Lines 124/125: "this method designed to tag the intermediary connectors of two non-interacting proteins."

Lines 147/149: "Subsequent steps involve utilizing streptavidin affinity immunoprecipitation and mass spectrometry proteomics to capture and identify the intermediary proteins"

To be precise, the method will tag proteins that are proximal to A (with pup(E)), one or some of them (protein(s) X) will also interact with C and those proteins that are proximal to X/C will be tagged with biotin, X will be part of it but among many others. Since the detection is solely based on biotin, the method cannot be specific to "intermediary proteins" (= those that got biotin and pup(E)).

This makes the experiment depicted on Fig. 2 more suggestive than conclusive. Fig. 2C suggests that AMOT will be the protein that undergoes first conjugation with Tb(N)-pup(E) and then activate Tb(C)-YAP1. While this would be logical and the data suggest it is what happens, this has not been fully demonstrated. Indeed another Protein X may be conjugated by Tb(N)-pup(E), then activate Tb(C)-YAP1 and lead to biotinylation of AMOT because it interacts with YAP1. The fact the AMOT runs slower on an SDS-PAGE upon transfection of PafA-β-actin suggests that it got conjugated with Tb(N)-pup(E) but this was not demonstrated.

If the authors really want to show their model holds, they may perform one of these control experiments:

- Immunoprecipitation with an anti-Flag antibody of the samples of Fig. 2F and immunoblot with anti-AMOT antibody: this will show that the size shift is indeed due to conjugation of Tb(N)-pup(E)

- STUPPIT like in Fig. 2E-G in cells in which AMOT was knocked down: if biotinylation and MS data are significantly changed this will show that AMOT was indeed the main intermediary protein

Without at least one of these two controls, if AMOT was identified because it interacted with actin prior it interacted with YAP1 is only supported by the current literature.

If done for approach I, the suggested control experiments of course do not need to be performed for approach II.

2) For all the western blots presented in this manuscript: inputs are missing a normalization control (total protein stain or housekeeping control)

3) Approach II is depicted on Fig. 3 and is in principle interesting, however the data on Fig. 3D are confusing:

* FLAG-blot, lanes 2 & 3: why is there no Tb(N)-pupE to see (it should be stably expressed)

* Streptavidin, lane 2: considerable biotinylation is observed with two fragments that are just floating around (Tb(C) and Tb(N)-pupE): why? Tb(C) & Tb(N) seems to self-assemble in the absence of actin-pafA. If true then the right negative control for STUPPIT is without pafA.

* Myc blots do not seem to make much sense for lanes (2) and (3): why should the size change, why should there be a signal at all when actin-pafA is not expressed?

* In any case, please include a full blot for Myc with all samples (like for Flag & Strep)

4) The data presented on Fig. 3d and 3f do not correlate.

5) Figure 4D: first blot should be IB:Flag (I guess) not IB: Streptavidin

6) Fig. 4C: If the data presented on Fig. 3D is right, the suitable control seems to be without Pap: how much reassembly would you already have with Tb(C) floating around?

7) For the set of data presented on Fig. 4 I have the same comment as for Fig. 2: I agree that STUPPIT identified α-catenin and LAPS1 but the only support for their identification as "intermediary proteins" is the current literature. The data that demonstrate they were conjugated with pup(E) is missing.

8) Line 293: what was the rational to select specifically α-catenin and SMAD4?

9) Fig. 5C: what are samples 1-6 and 7-12, replicate experiments?

10) Lines 293-304: The performed experiment is not expected to solely identify intermediary proteins of α-catenin and SMAD4 but proteins in close proximity to unknown proteins that previously interacted with SMAD4 and now in complex with α-catenin. Some of them may be intermediary proteins, others not. Therefore it seems at the first glance surprising that 5 out of 5 tested hits are all intermediary proteins. Was this expected, where the hits picked for validation chosen according to special criteria, does it suggest that α-catenin and SMAD4 are in fact part of the same complex?

Some more general comments:

- A pulldown with streptavidin is not an immunoprecipation (IP)

- Doing a t-test analysis based on 3 western blot experiments does not really make statistical sense: just show the individual data points for each replicate + a bar for the mean value (on all figures).

- The description of the MS analysis is not precise enough: MaxQuant yields intensity values, LFQs or iBAQs but does not perform enrichment calculations. Which software was use and with which parameters to perform imputations (if any), calculate fold changes and p-values. The MS data should be deposited in a public repository. These two points are absolutely necessary, especially for a method paper.

---

## [Decision Letter · Decision Letter 2]

31 Oct 2025

Dear Dr Ning,

Thank you for your patience while we considered your revised manuscript "A novel tool for labeling intermediary proteins between two non-interacting proteins" for publication as a Methods and Resources Article at PLOS Biology. This revised version of your manuscript has been evaluated by the PLOS Biology editors, the Academic Editor and the original reviewers.

Based on the reviews, I am pleased to say that we are likely to accept this manuscript for publication, provided you satisfactorily address the remaining points raised by Reviewer #2. This includes adding the data discussed in the rebuttal document (split-TurboID performed on Actin/AMOT vs Actin/YAP1) to the main manuscript and providing additional explanations/discussions as requested.

In addition, please make sure to address the following data and other policy-related requests that I have provided below (A-F):

(A) We routinely suggest changes to titles to ensure maximum accessibility for a broad, non-specialist readership. In this case, we would suggest a minor edit to the title, as follows. Please ensure you change both the manuscript file and the online submission system, as they need to match for final acceptance:

“STUPPIT is a proximity labeling tool for labeling intermediary proteins that bridge two non-interacting proteins””

(B) Please also ensure that each of the relevant figure legends in your manuscript include information on *WHERE THE UNDERLYING DATA CAN BE FOUND*, and ensure your supplemental data file/s has a legend.

(C) Thank you for already providing the raw and uncropped images for the Western blot data in S1_raw_images file. I have checked the file and this looks good, but I think some of the blots may be mislabeled? This includes Figure 2E, 2F, and S2C-D/S3B appear to be swapped around. In addition, please include the uncropped images for the figure discussed in the rebuttal to the SI file.

(D) Please ensure that your Data Statement in the submission system accurately describes where your data can be found and is in final format, as it will be published as written there. I reckon also including the accession number for the ProteomeXchange deposition in the statement (PXD069286).

(E) Per journal policy, if you have generated any custom code during the course of this investigation, please make it available without restrictions. Please ensure that the code is sufficiently well documented and reusable, and that your Data Statement in the Editorial Manager submission system accurately describes where your code can be found.

(F) Please note that per journal policy, the model system/species studied should be clearly stated in the abstract of your manuscript (e.g. human cell lines).

We expect to receive your revised manuscript within two weeks.

*Published Peer Review History*

*Press*

Best regards,

Richard

Richard Hodge, PhD

rhodge@plos.org

Reviewer remarks:

Reviewer #1: I appreciate that the authors have addressed all my concerns from the previous review. The revisions have substantially improved the clarity and rigor of the work, and I recommend acceptance of this manuscript.

Reviewer #2: I would like to congratulate the authors for the efforts they put in the revision of their manuscript which I found significantly improved. I have a few remaining minor redactional comments that I would recommend addressing to be fully supportive for publication.

Page 14 of the submission file:

"To experimentally verify this in our contex, we further generated two constructs AMOT-Flag-TbN and actin-Flag-TbN. Using these, we assessed the ligase self-labeling and split-TurboID (split at 73/74) activity under two conditions: AMOT-TbN co-expressed with TbC-YAP1, and actin-TbN co-expressed with TbC-YAP1. Consistent with the conclusions from (Cho et al., 2020) in their Fig. 3E, we observed that both the ligase self-biotinylation and TurboID enzymatic activity were far more robust—with significantly stronger signals—in the AMOT-TbN&TbC-YAP1 group than in the actin-TbN&TbC-YAP1 group. This result aligns with the known stronger interaction between AMOT and YAP1 (compared to actin and YAP1), further validating that split-TurboID (73/74) activity directly correlates with the strength of the interaction driving fragment proximity."

Please show and explain the figure corresponding to this experiment (split-TurboID performed on Actin/AMOT vs Actin/YAP1) in the manuscript. The figure can be placed in the supplementary material but it is an important control that provides an important back up to the claim that STUPPIT gives access to information that would not be available with split-turboID.

Response to comment #1

The experiment showed on S1B is good but it is not appropriately explained in the main text nor in the legend (one has to do a lot of guesswork to understand the setting). Please explicitly explain in the main text that the experiment shown in S1B corresponds to cells that were co-transfected with plasmids coding for actin-PafA-myc and AMOT-HA.

By contrast, the experiment shown in Fig. 2 corresponds to cells co-transfected with plasmids coding for actin-PafA-myc and HA-TbC-YAP1. Therefore I would also suggest writing in lines 155/159:

As expected, both endogenous (Fig. 2G) and exogenous AMOT (Fig. S1B) were successfully enriched following Flag-IP and exhibited molecular weight laddering detected by anti-AMOT and anti-HA antibody, in contrast to the no-transfection or the PafA-myc transfection control (Fig 2G and S1B Fig).

Response to comment #6

Thank you for performing and showing this control experiment in the revised manuscript. However you do not really explain in the main text why this experiment was performed and I think the general reader will need some help here. Therefore I would recommend adding in lanes 258/260 a few words to explain this control was needed because of the considerable background of split-TurboID (significant assembly of the two Tb fragments even when in the HA-TbC control).

In that vein I would also recommend showing (e.g. in the supplementary material) the MS data in AMOT-knockdown cells that you included in your response to comment #1 as this is another proof that AMOT itself served as one of the pupylated proteins that interacted with TbC-YAP1 and thereby reactivated TurboID activity.

Description of the analysis of the MS data:

There are still information missing: which software has been used to calculate enrichments and p-values. What do you define as p-value: for large data sets such as MS data, it is generally accepted that FDR (false discovery rate)-adjusted p-values (aka q-values) should be used. Was this the case here, if not, why?

Discussion:

I point two considerations the authors may add in their discussion to help potential users of their new method to optimize their experiments (I leave it to the authors if they want to add these or not):

1) Lines 386/387: one thing that one may consider with the choice of the split-system: both paf and TurboID attach their label (pupE and biotin) to lysine residues. It is therefore possible in certain cases that the first labeling step (pupIT) will saturate all available lysines and make the intermediate protein non-targetable by the second labeling step (biotinylation by TurboID). With this in mind the combination pupIT / split-APEX2 (with APEX2 rather biotinylating on tyrosines) may be worth trying.

2) In the discovery approach as shown on Fig. 5, identification of intermediate proteins is solely based on biotinylation. As discussed by the authors in lines 399/402, biotinylation generated by STUPPIT will label intermediate proteins and prey C but also proximity proteins between the intermediary protein and prey C. It may be worth mentioning that an even more targeted approach would be to first perform a FLAG-IP followed by streptavidin pulldown and MS analysis. This would allow specifically identifying "double labelled" proteins by STUPPIT, which would be even stronger candidates for intermediary proteins

As mentioned above I think these two considerations would be helpful for future potential users.

---

## [Editor Report · Decision Letter 3]

11 Nov 2025

Dear Dr Ning,

On behalf of my colleagues and the Academic Editor, Baojun Wang, I am pleased to say that we can accept your manuscript for publication, provided you address any remaining formatting and reporting issues. These will be detailed in an email you should receive within 2-3 business days from our colleagues in the journal operations team; no action is required from you until then. Please note that we will not be able to formally accept your manuscript and schedule it for publication until you have completed any requested changes.

PRESS

Best wishes,

Richard 

Richard Hodge, PhD

rhodge@plos.org

PLOS
